# Child marriage in rural Bangladesh and impact on obstetric complications and perinatal death: Findings from a health and demographic surveillance system

Kyu Han Lee[1], Atique Iqbal Chowdhury[2], Qazi Sadeq-ur Rahman[2], Solveig A. Cunningham[3], Shahana Parveen[4], Sanwarul Bari[2], Shams El Arifeen[2], Emily S. Gurley[1]*

1 Department of Epidemiology, Bloomberg School of Public Health, Johns Hopkins University, Baltimore, Maryland, United States of America, 2 Maternal and Child Health Division, icddr,b, Dhaka, Bangladesh, 3 Department of Global Health, Emory University, Atlanta, Georgia, United States of America, 4 Infectious Diseases Division, icddr,b, Dhaka, Bangladesh

* egurley1@jhu.edu

## Abstract

Adolescent pregnancies, a risk factor for obstetric complications and perinatal mortality, are driven by child marriage in many regions of South Asia. We used data collected between 2017–2019 from 56,155 married adolescents and women in a health and demographic surveillance system to present a population-level description of historical trends in child marriage from 1990–2019 as well as epidemiologic associations between maternal age and pregnancy outcomes in Baliakandi, a rural sub-district of Bangladesh. For pregnancies identified between 2017–2019, we used Kaplan-Meier estimates to examine timing of first pregnancies after first marriage and multinomial logistic regression to estimate associations between maternal age and perinatal death. We described the frequency of self-reported obstetric complications at labor and delivery by maternal age. In 1990, 71% of all marriages were to female residents under 18 years of age. This decreased to 57% in 2010, with the largest reduction among females aged 10–12 years (22% to 3%), and to 53% in 2019. Half of all newly married females were pregnant within a year of marriage, including adolescent brides. Although we observed a decline in child marriages since 1990, over half of all marriages in 2019 were to child brides in Baliakandi. In this same population, adolescent pregnancies were more likely to result in obstetric complications (13–15 years: 36%, 16–17 years: 32%, 18–34 years: 23%; $\chi^2$ test, p<0.001) and perinatal deaths (13–15 years: stillbirth OR 2.23, 95% CI 1.01–2.42; 16–17 years: early neonatal death OR 1.57, 95% CI: 1.01–2.42) compared to adult pregnancies. Preventing child marriage can improve the health of girls and contribute to Bangladesh's commitment to reducing child mortality.

**Data Availability Statement:** All data files are available from the Harvard Dataverse (https://doi.org/10.7910/DVN/93PYJC).

**Funding:** This work was supported by the Bill & Melinda Gates Foundation, Seattle, WA [award number OPP1126780]. The funder had no role in study design, data collection and analysis, decision to publish, or preparation of the manuscript.

**Competing interests:** The authors have declared that no competing interests exist.

## Introduction

Girls who become pregnant during adolescence are at increased risk for obstetric complications including pregnancy-induced hypertension, obstructed labor, obstetric fistula, and postpartum hemorrhage [1–4]. These complications are the leading cause of death among adolescent girls [5] and those who survive with conditions such as obstetric fistula experience devastating psychosocial disabilities spurred by shame, stigma, and rejection [6].

Adolescent pregnancies are also more likely to result in perinatal death [7], defined as fetal demise at or after 28 weeks of estimated gestational age (i.e. stillbirth [8]) or death within the first 7 days after live birth (i.e. early neonatal death [9]). While remarkable reductions in global under-five mortality have been observed over the past two decades, progress has been slower in preventing neonatal deaths, which made up nearly half of the estimated 5 million under-five deaths in 2019 [10]. Even less progress has been made in preventing stillbirths, which sums to an estimated 2 million fetal deaths each year [11].

Adolescent pregnancies are exacerbated by the practice of child marriage [12]. Child marriage, defined as the formal or informal union of a child under 18 years of age [13], is a human rights violation that disproportionately affects girls particularly in low- and middle-income countries [14], and contributes to severe social, developmental, and reproductive harms [15]. Globally, an estimated 16 million girls aged 15–19 become pregnant each year and 90% of these adolescent pregnancies occur within marriage [12]. Given societal and familial pressures to bear children immediately after marriage in many communities [15], marriage often means the beginning of a sexual relationship for children who are still in the process of maturing physically and psychologically [7]. Although extensive global efforts have been made to end child marriage, progress has been stagnant in recent years. In 2018, 21% of women aged 20–24 years worldwide were married before 18 years of age and, assuming the current rate of decline continues, it would take at least 50 years to end child marriage [14].

Some of the highest rates of child marriage in the world are found in Bangladesh. In 2018, 59% of women aged 20 to 24 years were married before 18 years of age and 28% of married adolescents aged 15 to 19 reported ever being pregnancy [16]. Much of our current understanding of child marriage prevalence in Bangladesh and the relationship with adverse pregnancy outcomes are derived from the Bangladesh Demographic and Health Survey, which utilizes a cross-sectional two-stage cluster sampling approach [16–18]. Although nationally representative, demographic indicators such as age at marriage may be underreported when collected retrospectively through cross-sectionally surveys, potentially to reduce dowry costs, which increase with the bride's age, and to meet the "preferred age" of brides, which is typically under 20 years [19, 20]. This type of misreported age may lead to overestimates of child marriage prevalence as well as introduce bias to epidemiologic associations between maternal age and adverse pregnancy outcomes.

In 2017, the Child Health and Mortality Prevention Surveillance (CHAMPS) network established a health and demographic surveillance system in Baliakandi, a rural sub-district of Bangladesh with approximately 216,000 residents [21]. We used data from nearly every resident in Baliakandi to observe how common child marriage was in this sub-district. Further, we conducted an in-depth investigation of the relationship between female age at marriage and timing of first pregnancies as well as the relationships between maternal age at delivery and key health outcomes such as obstetric complications and perinatal death.

## Methods

### Ethics statement

The study was approved by the icddr,b human subjects review committee (PR-16082). Written informed consent was obtained from adult participants and from married participants under 18 years of age.

### Inclusivity in global research

Additional information regarding the ethical, cultural, and scientific considerations specific to inclusivity in global research is included in S1 Checklist.

### Health and demographic surveillance system

We used data from an ongoing HDSS established by the Child Health and Mortality Prevention Surveillance (CHAMPS) network in Baliakandi, Bangladesh. All residents of Baliakandi sub-district were eligible for enrollment. Baseline surveys were conducted between March and August 2017 during which household and sociodemographic data were collected from every resident. Additional details on the Baliakandi HDSS are described by Cunningham et al. [21].

### Historic data

Married females under 50 years of age completed a birth history questionnaire either during baseline surveys or after in-migration. The questionnaire included questions regarding the date of marriage and prior pregnancies (S1 Table).

### Prospectively collected data

We conducted eleven rounds of household visits between September 2017 and February 2020. Frequent visits allowed for prompt identification of key demographic events such as marriages, pregnancies, pregnancy outcomes, deaths, and out-migrations. Pregnant residents provided the date of the last menstrual period, which was used as a proxy of pregnancy date and to estimate gestational age. If a resident was not present during a visit, demographic events missed during that round were later captured at subsequent household visits. We observed consent rates at or greater than 99% among all households in Baliakandi each year (approximately 52,000 households). These high consent rates indicate the HDSS captures demographic data and events from nearly every resident of Baliakandi. Given regular follow up, missing demographic data were uncommon for events included in this study.

### Age distribution of brides

For 1990 and 2016, we calculated the age distribution of brides using data from the birth history questionnaire (categorical age in years: $\leq 12$, 13–15, 16–17, and $\geq 18$). For 2017 to 2019, the annual proportion of marriages was estimated using marriages prospectively identified during HDSS household visits between September 2017 and December 2019. Data from the birth history questionnaire were used for in-migrants (S1 Fig).

### Time to pregnancy

We calculated the time between first marriage and pregnancy among female residents under 35 years of age and whose marriages were prospectively identified through the HDSS. Women 35 years and older were excluded as marriages were rare for this age group. As newly married brides often out-migrated with their husbands soon after marriage, we restricted the analysis

to individuals who remained a resident of Baliakandi for at least 180 days after marriage (S1 Fig). We used Kaplan-Meier estimates to examine the time it took for a female resident to become pregnant after their first marriage and whether this relationship differed by age at marriage (categorical age in years: 12–15, 16–17, and 18–34).

### Perinatal death

We used multinomial logistic regression to estimate the association between maternal age at delivery (age category in years: 13–15, 16–17, and 18–34) and stillbirths and early neonatal deaths among all singleton births that occurred between September 2017 and August 2019 (S2 Fig). Mothers aged 35 and older were excluded as births were rare for this group and advanced maternal age is a known risk factor of perinatal deaths [22]. We defined stillbirth as fetal demise at or after 28 weeks of estimated gestational age and early neonatal death as death within the first 7 days after live birth. To consider the influence of nulliparity and household wealth, we repeated the model after restricting the analysis to nulliparous pregnancies and adjusting for household wealth (categorical quintile based on the Demographic and Health Survey wealth index score [23]).

### Obstetric complications during labor and delivery

In December 2018, we supplemented the Baliakandi HDSS with additional surveys that identified adverse events during the pregnancy and up to 48 hours after delivery (S2 Table). All pregnant women identified during HDSS household visits were contacted within 3 days of giving birth and a survey on self-reported obstetric complications during labor and delivery was conducted by staff trained to describe these complications in lay language. We calculated the frequency of complications—prolonged/obstructed labor/failure to progress, birth trauma or difficult delivery, high blood pressure, heavy bleeding during delivery, severe headaches with blurred vision, fetal malpresentation, high fever with abdominal pain, high fever with smelly discharge, unplanned hospital admission for delivery (planned to deliver outside a hospital), and preterm birth—by age group for all births between January and August 2019 (S2 Fig). Preterm birth was defined as a birth before an estimated gestational age of 37 weeks. Chi-squared test and Fisher's exact test were used to test for statistical differences in the frequency of complications by maternal age.

## Results

### Age at marriages among female residents

A total 56,155 female residents of Baliakandi were married between 1990 and 2019; all provided age at marriage. Among female residents married in 1990, 71% were married before 18 years of age (915/1,284) (Fig 1). This proportion dropped to 57% among female residents married in 2010 (1,185/2,063). Between 1990 and 2010, the proportion of marriages to females under 13 years decreased from 22% (288/1,284) to 3% (72/2,063), females aged 13–15 years decreased from 32% (411/1,284) to 26% (538/2,063) and females aged 16–17 years increased from 17% (216/1,284) to 28% (575/2,063). The mean age at marriage increased from 15.3 years (standard deviation (SD) 3.2) in 1990 to 17.3 years (SD 3.4) in 2010.

Between 2010 and 2019, the overall proportion of marriages to female residents under 18 years dropped from 57% (1,185/2,063) to 53% (1,698/3,210). The proportion to females under 13 years decreased from 3% (72/2,063) to 1% (28/3,210), females aged 13–15 years decreased from 26% (538/2,063) to 21% (671/3,210), females aged 16–17 years increased from 28% (575/

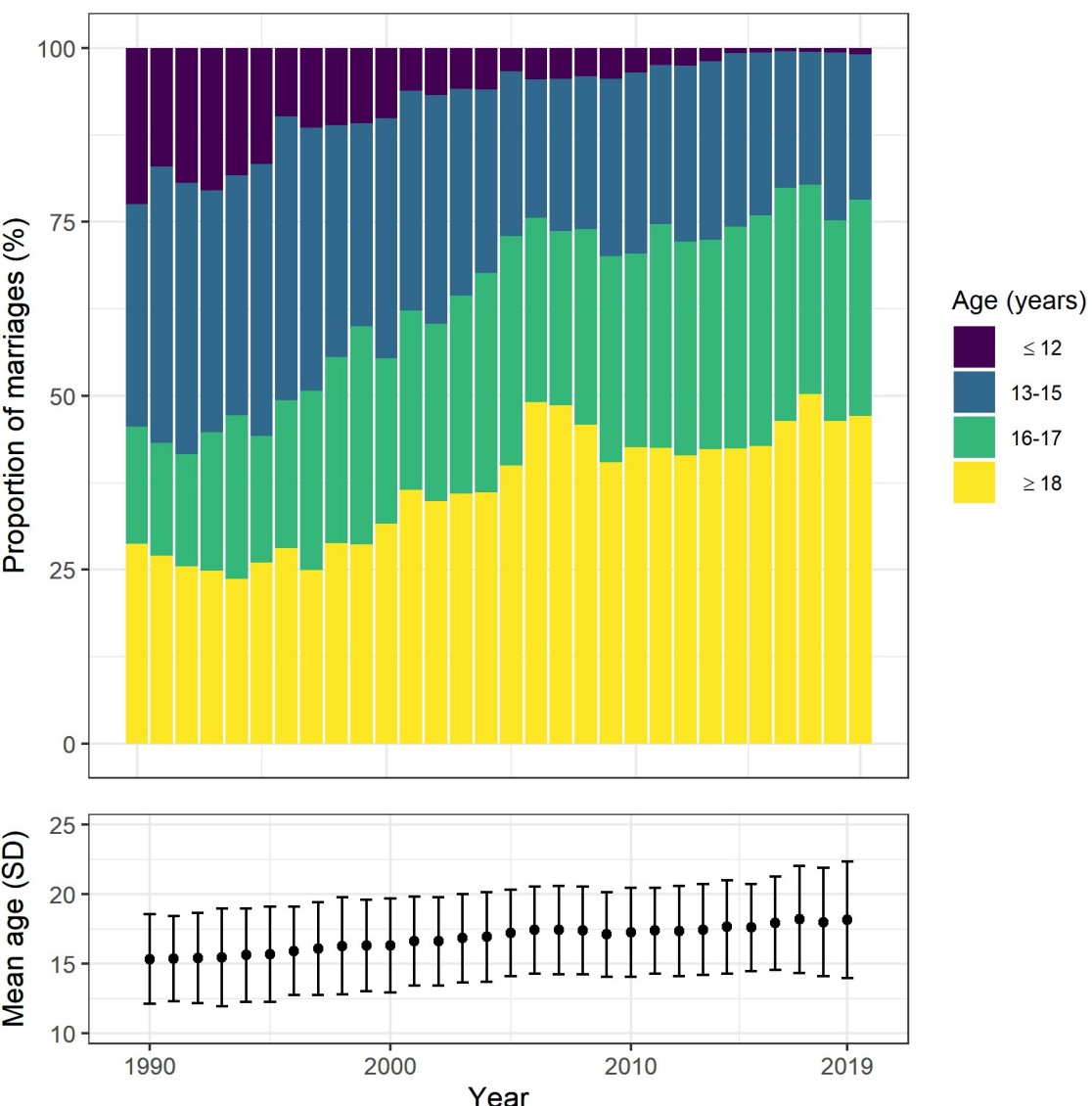

**Fig 1. Age at marriage among 56,155 female residents of Baliakandi sub-district, Bangladesh, 1990 to 2019.** Bars represent the annual proportion of marriages to specific age groups. Points and error bars represent the mean age at marriage and standard deviation.

2,063) to 31% (999/3,210). The mean age at marriage increased from 17.3 years (SD 3.4) in 2010 to 18.2 years (SD 4.2) in 2019.

## Characteristics of first marriages

A total of 3,764 female residents under 35 years of age reported first marriages in Baliakandi between September 2017 and August 2019 (S3 Table); household wealth was available for all but one and age of the groom was available for all but one. Younger brides were more likely to be in the lowest household wealth quintile: 25% among ages 10–15 years (271/1,077), 20% among ages 16–17 years (227/1,150), and 15% among females ages 18–34 years (236/1,537) ($\chi^2$ test, p<0.001). Among all first marriages, males were a median of 8 years (interquartile range

(IQR) 5–10) older than their female spouses. The median age difference was 9.5 years (IQR 8–12) for females married before 13 years of age, 9 years (IQR 6 to 12) for those married at ages 13–15 years, 8 years (IQR 6–10) for those married at ages 16–17 years, and 6 years (IQR 3–9) among those married at ages 18–34 years.

### Time between first marriage and pregnancy

Approximately half of all newly married females who remained in Baliakandi for 180 days (N = 1,320) had a pregnancy within 365 days of marriage; Kaplan-Meier estimates: 52% (95% confidence interval (CI) 47–57%) among those married at 12–15 years, 53% (95% CI 48–59%) among those married at ages 16–17 years and 50% (95% CI 45–54%) among those married at ages 18–34 years (Fig 2). We found no statistically significant difference by age at first marriage (log-rank test, p = 0.535).

### Maternal age at birth and perinatal mortality

We identified a total 8,806 singleton births among female residents under 35 years of age between September 2017 and August 2019. A total of 198 births were stillborn and 188 live births resulted in early neonatal death (Table 1). We observed a stillbirth rate of 22 stillbirths per 1,000 singleton births and early neonatal mortality rate of 22 deaths per 1,000 singleton live births. Sixteen percent of perinatal deaths were among mothers under 18 years of age (61/386) and 52% were among first-time mothers (202/386). We did not observe any singleton births among females under 13 years of age during this period.

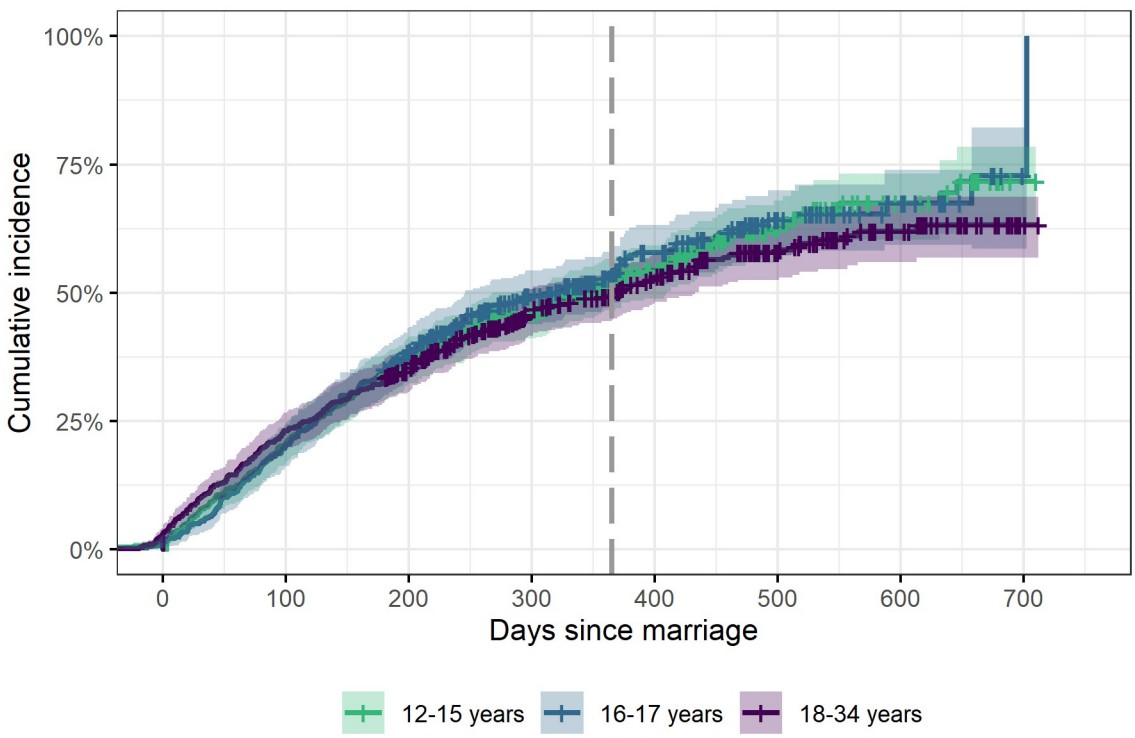

**Fig 2. Cumulative incidence of pregnancies after first marriage and 95% confidence intervals, by age at marriage.** Among 1,320 females under 35 years of age who resided in Baliakandi for at least 180 days after marriage, Baliakandi sub-district, Bangladesh. Grey dashed line indicates 365 days after marriage.

**Table 1. Characteristics of 8,806 singleton births among females under 35 years of age, Baliakandi sub-district, Bangladesh, September 2017 to August 2019.**

| Characteristic | All births N = 8,806 n (%) | Live births surviving 7 days N = 8,420 n (%) | Stillbirths N = 198 n (%) Rate | Early neonatal deaths N = 188 n (%) Rate |
|---|---|---|---|---|
| Maternal age at delivery (years)* | | | | |
| 13 to 15 | 238 (3%) | 221 (3%) | 11 (6%) 46 per 1,000 births | 6 (3%) 26 per 1,000 live births |
| 16 to 17 | 769 (9%) | 725 (9%) | 20 (10%) 26 per 1,000 births | 24 (13%) 32 per 1,000 live births |
| 18 to 29 | 6,601 (75%) | 6338 (75%) | 135 (68%) 20 per 1,000 births | 128 (68%) 20 per 1,000 live births |
| 30 to 34 | 1,198 (14%) | 1136 (13%) | 32 (16%) 27 per 1,000 births | 30 (16%) 26 per 1,000 live births |
| Parity | | | | |
| Nulliparous | 3,827 (43%) | 3,625 (43%) | 103 (52%) | 99 (53%) |
| 1 | 3,252 (37%) | 3,146 (37%) | 57 (29%) | 49 (26%) |
| 2 | 1,315 (15%) | 1,261 (15%) | 26 (13%) | 28 (15%) |
| 3+ | 412 (5%) | 388 (5%) | 12 (6%) | 12 (6%) |
| Household wealth quintile | | | | |
| Highest | 1,815 (21%) | 1,751 (21%) | 28 (14%) | 36 (20%) |
| High | 1,863 (21%) | 1,786 (21%) | 45 (23%) | 32 (22%) |
| Middle | 1,809 (21%) | 1,727 (21%) | 42 (21%) | 40 (21%) |
| Low | 1,689 (19%) | 1,610 (19%) | 37 (19%) | 42 (17%) |

*No singleton births were reported among mothers under 12 years of age. Births among women over 34 years were excluded.

Compared to mothers ages 18–34 years, mothers ages 13–15 years had higher odds of stillbirth (OR 2.23, 95% CI 1.19–4.16) and females ages 16–17 years had higher odds of early neonatal death (OR 1.57, 95% CI 1.01–2.42). Among nulliparous mothers (N = 3,827), the association between ages 13–15 years and stillbirth was not statistically significant (adjusted odds ratio (aOR) 1.77, 95% CI 0.92–3.41) and the association between ages 16–17 years and early neonatal death was not statistically significant (aOR 1.28, 95% CI 0.41–2.04), after adjusting for household wealth (S4 Table). Nulliparous mothers in the two highest household wealth quintiles had lower odds of early neonatal deaths (highest: aOR 0.52, 95% CI 0.27–0.97; high: aOR 0.46, 95% CI 0.24–0.88).

## Obstetric complications during labor and delivery

Information on self-reported obstetric complications during labor and delivery were collected for 3,286 singleton births (99% of total 3,311 singleton births) between January and August 2019 among female residents who were under 35 years of age at time of birth. Complications were common in Baliakandi regardless of maternal age, with 24% of all births (803/3,286) reporting at least one obstetric complication, including prolonged/obstructed labor or failure to progress, birth trauma or difficult delivery, and heavy bleeding (Table 2). Nineteen percent (610/3,286) involved unplanned hospital admissions. Based on HDSS data, 19% of live births (604/3,223) were considered preterm according to estimated gestational age and 38% (180/604) of preterm births involved pre-labor cesarean sections. Fifty-two percent (1662/3221) of singleton births involved a cesarean section and vacuum extraction was not reported in any vaginal deliveries. Maternal deaths were infrequent. We identified a total seven deaths among all female residents who had any type of pregnancy outcome (miscarriage or birth) between January and August 2019 (n = 3912). All deaths were among women 19 to 35 years of age who

**Table 2. Self-reported obstetric complications during labor and delivery among 3,286 singleton births by female residents under 35 years of age, by maternal age at birth, Baliakandi sub-district, Bangladesh, January to August 2019.**

| Events | All N = 3,286 n (%) | 13 to 15* years N = 101 n (%) | 16 to 17 years N = 291 n (%) | 18 to 34 years N = 2,894 n (%) | P-value[†] |
|---|---|---|---|---|---|
| Any complications[‡] | 803 (24%) | 36 (36%) | 93 (32%) | 674 (23%) | <0.001 |
| Prolonged/obstructed labor/failure to progress | 267 (8%) | 14 (14%) | 33 (11%) | 220 (8%) | 0.008 |
| Birth trauma or difficult delivery | 276 (8%) | 17 (17%) | 42 (14%) | 217 (8%) | <0.001 |
| High blood pressure | 145 (5%) | 4 (4%) | 15 (5%) | 126 (5%) | 0.728 |
| Heavy bleeding during delivery | 120 (4%) | 9 (9%) | 10 (3%) | 101 (4%) | 0.029 |
| Severe headaches with blurred visions | 114 (3%) | 5 (5%) | 11 (4%) | 98 (3% | 0.550 |
| Fetal malpresentation | 39 (1%) | 2 (2%) | 2 (1%) | 35 (1%) | 0.456 |
| High fever with abdominal pain | 29 (1%) | 1 (1%) | 1 (0%) | 27 (1%) | 0.548 |
| High fever with smelly discharge | 19 (1%) | 1 (1%) | 0 (0%) | 18 (1%) | 0.297 |
| Unplanned hospital admission for delivery | 610 (19%) | 28 (28%) | 79 (27%) | 503 (17%) | <0.001 |
| Preterm birth[•] | 604/3,223 (19%) | 14/98 (14%) | 51/287 (18%) | 539/2838 (19%) | 0.456 |

*No singleton birth among children under 13 years of age with data on obstetric complications.

[†]Chi-squared test or Fisher's exact test; comparing all three age groups

[‡]Includes any of the complications listed below

[•]Only among live births

had a singleton live birth. Six deaths were likely related to pregnancy, occurring within a month of birth. One death occurred approximately 9 months after birth.

Compared to females ages 18–34 years, adolescents were more likely to report at least one complication (13–15 years: 36%, 16–17 years: 32%, 18–34 years: 23%; $\chi^2$ test, p<0.001) and have unplanned admissions (13–15 years: 28%, 16–17 years: 27%, 18–34 years: 17%; $\chi^2$ test, p<0.001). Among specific complications, prolonged/obstructed labor or failure to progress (13–15 years: 14%, 16–17 years: 11%, 18–34 years: 8%), birth trauma or difficult delivery (13–15 years: 17%, 16–17 years: 14%, 18–34 years: 8%), and heavy bleeding were more common among females under 18 years of age (13–15 years: 9%, 16–17 years: 3%, 18–34 years: 4%), compared to ages 18–34 years ($\chi^2$ test or Fisher's exact test, all p<0.050).

## Discussion

In this study, we examined how common child marriage was in Baliakandi, rural region of Bangladesh, and investigated the relationship between child marriage, adolescent pregnancies, and adverse pregnancy outcomes such as perinatal death and obstetric complications. Using data from a HDSS covering approximately 216,000 residents in a rural sub-district of Bangladesh, we found that more than half of all marriages in Baliakandi were to child brides in 2019, a number that has remained unchanged from the prior decade. Despite differences in study design, our findings were consistent with national trends described in the 2017–2018 Bangladesh Demographic Health Survey [16] and highlights the need to accelerate ongoing commitments to end child marriage and promote additional widespread interventions.

In Baliakandi, approximately half of all newly married female residents, including adolescents, became pregnant within a year of marriage. Although childbearing should be delayed among adolescent girls to protect the health of the mother and newborn [7, 15, 24], our study showed no evidence of postponed pregnancies. These findings are consistent with earlier studies which showed the frequency of early fertilization (birth within the first year of marriage)

did not differ significantly by age at marriage in Bangladesh, India, Nepal, and Pakistan [18, 25]. Adolescent pregnancies are likely driven by the social value placed on childbearing for wives in Bangladesh and the prospect of improving one's status within the husband's family through childbirth [26, 27]. Many newly married adolescents lack power and social status to negotiate delays in pregnancy, coupled with a lack of knowledge or misconceptions of contraceptives. One example of misconceptions is that contraceptives can lead to infertility [27]. In contrast, wives who delay childbearing often experience stigma of perceived infertility, abuse by in-laws, and rumors of infidelity [26]. These dynamics may drive low contraceptive use among adolescent brides [16].

Adolescents who become pregnant are more likely to experience obstetric complications such as prolonged and obstructed labor, which can cause long-term physical consequences [6, 28, 29] as well as devastating psychosocial disabilities even after treatment, as girls deal with shame, stigma, and rejection by their families and communities [6]. In Baliakandi, obstetric complications were twice as likely for pregnant adolescents compared to adults. A quarter of deliveries among adolescent girls involved an unplanned hospital admission. A third of reported complications included prolonged or obstructed labor, birth trauma or difficult deliveries, and heavy bleeding, which are related to physical immaturity [7] and can lead to severe, long-term urologic, gynecologic and neurologic injuries as well as secondary infertility [6, 28, 29]. In addition to complications, adolescent girls who became pregnant were approximately twice as likely to have a stillbirth or for their babies to die in the first few days of life than adult women, consistent with associations reported in the 2017–2018 Bangladesh Demographic and Health Survey [30, 31]. The high risk of perinatal death could partially be explained by adolescents being more likely to be nulliparous, a risk factor for obstetric complications and perinatal death in Bangladesh [31, 32]. Associations between adolescence and perinatal outcomes are less consistent in earlier studies in South Asia but this may be due to differences in study population and how maternal age was categorized [33, 34].

Our findings highlight the role of poverty in both child marriage and perinatal death. In Bangladesh, girls are often viewed as financial burdens for the family and dowries paid to the groom's family are typically smaller for younger brides [35]. In addition, perinatal deaths are more likely to occur in less wealthy households, perhaps due to poor nutrition, lack of resources, and less access of health care [36–39]. The preterm birth rate in Baliakandi (19% of all singleton live births) was consistent with national estimates, which is one of the highest in the world [40]. Preterm births may be common due to the regular practice of elective cesarean sections in Baliakandi which may occur without medical indication due to a fear of vaginal deliveries and a general belief that cesarean deliveries are safer for the mother and baby [41, 42].

A notable strength of this study was the use of prospectively collected HDSS data to investigate child marriage. Nearly all estimates of child marriage in Bangladesh and epidemiologic investigations of adolescent pregnancies and adverse pregnancy outcomes are based on the Bangladesh Demographic and Health Survey [16, 17, 30]. The Baliakandi HDSS allows us to piece together a detailed picture of maternal and child health over time—at marriage, during pregnancies and delivery, and as a mother. Although Baliakandi represents a small portion of rural Bangladesh, our findings may reflect circumstances in other rural regions of the country given regional similarities in child marriage prevalence, adolescent childbearing, and age at first birth [14]. Our study did not include female residents who out-migrated nor those who died. It is unclear how these deaths might impact our estimates. Further, obstetric complications were self-reported. To minimize potential reporting bias, obstetric complications were described in lay language and questions were asked by data collectors trained to explain these complications.

The problem of child marriage is daunting; it is a practice deeply rooted in social and gender norms and sustained by gender inequality, poverty and insecurity [15, 35, 43, 44]. These relationships caution a likely increase in rates of child marriage during the COVID-19 pandemic due to school closures and increased social insecurities [45]. Given these challenges, we recognize the notable increase in mean age among child brides in Baliakandi and the promising global and national efforts to end child marriage [14, 46]. Findings from a cluster randomized controlled trial in Bangladesh show interventions such as education support for adolescents, gender rights awareness, and livelihood training can lead to a 25% to 30% reduction in child marriage [46]. Only by accelerating ongoing commitments and promoting additional widespread interventions we can truly eliminate child marriage. By addressing child marriage, we improve the health of two groups of children: adolescent girls and infants. We can prevent a chain of harmful social, developmental, and reproductive harms that have lasting impacts on adolescent girls while reducing the global burden of stillbirths and neonatal deaths.

## Conclusion

Our study shows over half of all marriages in Baliakandi, Bangladesh, were to child brides in 2019. In this community, early marriage contributes to adolescent pregnancies, a risk factor for perinatal death and obstetric complications. These findings highlight the need to accelerate national and global commitments to end child marriage, protect the health of adolescent girls and, reduce the burden of perinatal mortality in Bangladesh.

## Supporting information

**S1 Fig. Baliakandi health and demographic surveillance system data and analyses used to estimate proportion of marriage to female children and time between first marriage and pregnancy in Baliakandi, Bangladesh.**
(TIF)

**S2 Fig. Baliakandi health and demographic surveillance system data and analyses used to examine relation between maternal age at birth, complications during delivery, and perinatal mortality in Baliakandi, Bangladesh.**
(TIF)

**S1 Table. Excerpt from birth history questionnaire.**
(DOCX)

**S2 Table. Excerpt from pregnancy surveillance questionnaire.**
(DOCX)

**S3 Table. Characteristics of female residents under 35 years of age who had first marriages, Baliakandi sub-district, Bangladesh, September 2017 to August 2019.**
(DOCX)

**S4 Table. Crude and adjusted odds ratios of stillbirth and early neonatal death, compared to live births surviving more than 7 days.** Adjusted model restricted to nulliparous mothers and adjusted for maternal age and household wealth. Total 8,806 singleton births among female residents under 35 years at birth, Baliakandi sub-district, Bangladesh, September 2017 to August 2019.
(DOCX)

**S1 Checklist. Inclusivity in global research.**
(DOCX)

## Author Contributions

**Conceptualization:** Kyu Han Lee, Emily S. Gurley.

**Data curation:** Atique Iqbal Chowdhury, Qazi Sadeq-ur Rahman.

**Formal analysis:** Kyu Han Lee.

**Funding acquisition:** Sanwarul Bari, Shams El Arifeen, Emily S. Gurley.

**Investigation:** Kyu Han Lee, Atique Iqbal Chowdhury, Shahana Parveen.

**Methodology:** Kyu Han Lee, Solveig A. Cunningham, Emily S. Gurley.

**Project administration:** Sanwarul Bari.

**Software:** Qazi Sadeq-ur Rahman.

**Supervision:** Shams El Arifeen, Emily S. Gurley.

**Visualization:** Kyu Han Lee.

**Writing – original draft:** Kyu Han Lee.

**Writing – review & editing:** Atique Iqbal Chowdhury, Qazi Sadeq-ur Rahman, Solveig A. Cunningham, Shahana Parveen, Sanwarul Bari, Shams El Arifeen, Emily S. Gurley.

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
