## [Decision Letter · Decision Letter 0]

23 Nov 2022

PONE-D-22-26467Child marriage in rural Bangladesh and impact on obstetric complications and perinatal death: findings from a health and demographic surveillance systemPLOS ONE

Dear Dr. Gurley,

Thank you for submitting your manuscript to PLOS ONE. After careful consideration, we feel that it has merit but does not fully meet PLOS ONE’s publication criteria as it currently stands. Therefore, we invite you to submit a revised version of the manuscript that addresses the points raised during the review process.

We look forward to receiving your revised manuscript.

Kind regards,

Forough Mortazavi

Academic Editor

PLOS ONE

Journal Requirements:

3. You indicated that you had ethical approval for your study. Please clarify whether minors (participants under the age of 18 years) were included in this study. If yes, in your Methods section, please ensure you have also stated whether you obtained consent from parents or guardians of the minors included in the study or whether the research ethics committee or IRB specifically waived the need for their consent.

4. Please include your ethics statement in the Methods section of your manuscript. In the Methods section of your revised manuscript, please include the full name of the institutional review board or ethics committee that approved the protocol, the approval or permit number that was issued, and the date that approval was granted.

Additional Editor Comments:

Dear authors,

Thank you for working on this topic. Please clarify how women especially uneducated ones could report difficult to diagnose birth complications such as labor/failure to progress, birth trauma or difficult delivery, high blood pressure, heavy bleeding during delivery, fetal malpresentation, high fever with abdominal pain, and high fever with smelly discharge,? How reliable are the collected data?

Reviewers' comments:

Reviewer's Responses to Questions

**Comments to the Author**

1. Is the manuscript technically sound, and do the data support the conclusions?

Reviewer #1: Partly

2. Has the statistical analysis been performed appropriately and rigorously? 

Reviewer #1: Yes

3. Have the authors made all data underlying the findings in their manuscript fully available?

Reviewer #1: Yes

4. Is the manuscript presented in an intelligible fashion and written in standard English?

Reviewer #1: Yes

5. Review Comments to the Author

Reviewer #1: Child marriage in rural Bangladesh and impact on obstetric complications and perinatal death: findings from a health and demographic surveillance system

Thank you for providing this opportunity for me to review this manuscript. Please see my comments as follows:

Abstract:

1. Please report how many women were assessed?

2. And please report timeline of the study.

Methods

1. Please mention the design of the study.

2. Please provide reference for stillbirth, preterm birth and early neonatal death.

3. Please write the inclusion/exclusion criteria.

4. Overall, please re-arrange the manuscript according to STORBE guideline.

5. As some maternal and neonatal deaths are contributed to place of birth (home or hospital), birth attendant (skilled midwife or other health provider), or the number of prenatal care, there are need more information such as did adolescents delivered in hospital, and who was their birth attendants? skilled midwife or nurse? These information should be controlled when you report maternal or neonatal death.

Results

1. What was the ratio of vaginal delivery to cesarean section and also what was the rate of vaginal delivery using instrument such as vacuum?

2. Please put some information such as educational level of participants and their husbands, household member in demographic table.

3. Please report maternal death and its cause.

4. Table 2: Authors mention severe headache with blurred vision. Did they mean preeclampsia? If so, please use its correct term.

5. Again, for high fever and smelly discharge, did they mean postpartum infection?

6. What authors mean about un-planned hospital delivery? Is that mean women delivered at home and only some of them go to hospital?

7. Please provide a table for multiple logistic regression, until readers to be able to see what confounders were controlled.

Discussion

1. Please mention the objective/s of the study at the beginning of the discussion.

2. Discussion needs more comparison between the findings of the study and results of other studies.

3. What was the limitations of the study?

4. What was the conclusion according to the objectives of the study?

6. PLOS authors have the option to publish the peer review history of their article (what does this mean?). If published, this will include your full peer review and any attached files.

Reviewer #1: **Yes: **Parvin Abedi

---

## [Author Response · Author response to Decision Letter 0]

20 Dec 2022

Response to journal requirements:

The manuscript has been revised to match the PLOS ONE style templates.

The questionnaire has been completed and uploaded as Supporting Information S1 Checklist.

3. You indicated that you had ethical approval for your study. Please clarify whether minors (participants under the age of 18 years) were included in this study. If yes, in your Methods section, please ensure you have also stated whether you obtained consent from parents or guardians of the minors included in the study or whether the research ethics committee or IRB specifically waived the need for their consent.

Married minors are included in this study. Written informed consent was directly obtained from these participants. This clarification has been added to the ethics statement, now in the Methods section (lines 83-86)

4. Please include your ethics statement in the Methods section of your manuscript. In the Methods section of your revised manuscript, please include the full name of the institutional review board or ethics committee that approved the protocol, the approval or permit number that was issued, and the date that approval was granted.

The ethics statement has been added to the Methods section (lines 83-86).

Response to Editor Comments

Thank you for working on this topic. Please clarify how women especially uneducated ones could report difficult to diagnose birth complications such as labor/failure to progress, birth trauma or difficult delivery, high blood pressure, heavy bleeding during delivery, fetal malpresentation, high fever with abdominal pain, and high fever with smelly discharge,? How reliable are the collected data?

Thank you for your comment. Complications were self-reported. To minimize potential reporting bias, we used common language rather than clinical definitions (e.g. high fever with abdominal pain) or included multiple terms that may represent the same conditions (e.g. prolonged labor, obstructed labor, or failure to progress). All questions were asked by data collectors who were trained to thoroughly describe these conditions. We included a clarification in line 151 and added a limitation statement in the discussion (lines 321-323). If there is a relationship between reporting and education, we would expect bias towards the null, with fewer complications reported among less educated adolescents.

Response to Review Comments

Reviewer #1: Child marriage in rural Bangladesh and impact on obstetric complications and perinatal death: findings from a health and demographic surveillance system

Thank you for providing this opportunity for me to review this manuscript. Please see my comments as follows:

Thank you for your insightful comments. Please see our responses below.

Abstract:

1. Please report how many women were assessed?

The total number was added in line 20.

2. And please report timeline of the study.

The study years were added in line 20.

Methods

1. Please mention the design of the study.

The study design is a health and demographic surveillance system. It is included in the title and in the Methods section (lines 93-94). 

2. Please provide reference for stillbirth, preterm birth and early neonatal death.

References were added to lines 44-46.

3. Please write the inclusion/exclusion criteria.

Data were from a health and demographic surveillance system with the only inclusion criterion being a resident of the catchment area. We have added a clarifying statement in lines 94-95.

4. Overall, please re-arrange the manuscript according to STORBE guideline.

STROBE does not recommend a specific arrangement. According to the original article describing STROBE (Elm 2007), “the order and format for presenting information depends on author preferences, journal style, and the traditions of the research field.” If there are elements in reporting that the reviewer feels are missing, we are happy to address those.

Reference: Elm et al. 2007. The Strengthening the Reporting of Observational Studies in Epidemiology (STROBE) statement: guidelines for reporting observational studies. https://www.thelancet.com/journals/lancet/article/PIIS0140-6736(07)61602-X/fulltext#article_upsell 

5. As some maternal and neonatal deaths are contributed to place of birth (home or hospital), birth attendant (skilled midwife or other health provider), or the number of prenatal care, there are need more information such as did adolescents delivered in hospital, and who was their birth attendants? skilled midwife or nurse? These information should be controlled when you report maternal or neonatal death.

Thank you for your thoughtful comment. We consider these factors to be mediators between adolescence and perinatal deaths, rather than confounders. We understand these mediators are critical for maternal health. However, for this study, we wanted to focus on what the public health community could do at the beginning of the causal chain, which in this case is child marriage, rather than untangling potential mediators of this relationship.

Results

1. What was the ratio of vaginal delivery to cesarean section and also what was the rate of vaginal delivery using instrument such as vacuum?

We’ve included the proportion of all singleton births with cesarean section and the proportion of all vaginal deliveries involving vacuum extraction in lines 240-242.

2. Please put some information such as educational level of participants and their husbands, household member in demographic table.

To capture socio-economic indicators of study participants, we used a household wealth index instead, which is included in the demographic table (Table 1). Education level was not considered an optimal indicator as the upper bound would be limited by age (e.g., mothers aged 12-15 would only have partial secondary education), and would be less useful for comparing groups. 

3. Please report maternal death and its cause.

We identified a total 7 deaths among residents who had any pregnancy outcome (miscarriage or birth) between January and August 2019. All deaths were among women 19 to 35 years of age who had a singleton live birth. Six deaths were likely related to pregnancy, occurring within a month of birth. One death occurred approximately 9 months after birth. These results were added to lines 242-246. Unfortunately, the cause of maternal death was not systematically investigated.

4. Table 2: Authors mention severe headache with blurred vision. Did they mean preeclampsia? If so, please use its correct term.

Complications were self-reported in a population that may not be familiar with clinical definitions such as “preeclampsia”. To minimize potential reporting bias, we used common language rather than clinical definitions (e.g. high fever with abdominal pain) or included multiple terms that may represent the same conditions (e.g. prolonged labor, obstructed labor, or failure to progress). All questions were asked by data collectors who were trained to describe these conditions. We included a clarification in lines 151 and added a limitation statement in the discussion (lines 321-323).

5. Again, for high fever and smelly discharge, did they mean postpartum infection?

Please see response to #4 above.

6. What authors mean about un-planned hospital delivery? Is that mean women delivered at home and only some of them go to hospital?

Yes, this meant the mother planned to deliver at home but ended up at the hospital for delivery. A clarification was added to lines 155.

7. Please provide a table for multiple logistic regression, until readers to be able to see what confounders were controlled.

This information is available as Table S4.

Discussion

1. Please mention the objective/s of the study at the beginning of the discussion.

We have included the objectives of the study as requested.

2. Discussion needs more comparison between the findings of the study and results of other studies.

We expanded our literature search to other studies including those in South Asia [lines 277-279, 302-304].

3. What was the limitations of the study?

We believe the main limitations of this study is 1) the role of out-migration and deaths among female residents (lines 319-320) and the self-reported nature of complications. We’ve added the second point in lines 320-323 in response to your earlier comment.

4. What was the conclusion according to the objectives of the study?

A conclusion section was added (lines 339-343).

---

## [Decision Letter · Decision Letter 1]

7 May 2023

PONE-D-22-26467R1Child marriage in rural Bangladesh and impact on obstetric complications and perinatal death: findings from a health and demographic surveillance systemPLOS ONE

Dear Dr. Emily Gurley

Thank you for submitting your manuscript to PLOS ONE. After careful consideration, we feel that it has merit .

There is one minor comment by Reviewer 2 that I would like you to invite before we proceed 

We look forward to receiving your revised manuscript.

Kind regards,

Fadhlun Alwy Al-beity, MMed, PhD (ongoing)

Academic Editor

PLOS ONE

Journal Requirements:

Reviewers' comments:

Reviewer's Responses to Questions

**Comments to the Author**

1. If the authors have adequately addressed your comments raised in a previous round of review and you feel that this manuscript is now acceptable for publication, you may indicate that here to bypass the “Comments to the Author” section, enter your conflict of interest statement in the “Confidential to Editor” section, and submit your "Accept" recommendation.

Reviewer #1: All comments have been addressed

Reviewer #2: All comments have been addressed

2. Is the manuscript technically sound, and do the data support the conclusions?

Reviewer #1: Yes

Reviewer #2: Yes

3. Has the statistical analysis been performed appropriately and rigorously? 

Reviewer #1: Yes

Reviewer #2: Yes

4. Have the authors made all data underlying the findings in their manuscript fully available?

Reviewer #1: Yes

Reviewer #2: Yes

5. Is the manuscript presented in an intelligible fashion and written in standard English?

Reviewer #1: Yes

Reviewer #2: Yes

6. Review Comments to the Author

Reviewer #1: Thank authors as they responded all of my comments, and the manuscript is not ready for publication.

Reviewer #2: The authors have addressed all comments well. While I have no concerns and its not necessary to make any edits, the one thing that might be helpful would be to provide an assessment in the discussion of how generalizable these results from a fairly small rural sub-district might be to all of rural Bangladesh.

7. PLOS authors have the option to publish the peer review history of their article (what does this mean?). If published, this will include your full peer review and any attached files.

Reviewer #1: No

Reviewer #2: **Yes: **Russell S. Kirby

---

## [Author Response · Author response to Decision Letter 1]

22 May 2023

We thank the reviewers for their feedback and comments. Please find our response to the comment listed by Reviewer #2.

Reviewer #2: The authors have addressed all comments well. While I have no concerns and its not necessary to make any edits, the one thing that might be helpful would be to provide an assessment in the discussion of how generalizable these results from a fairly small rural sub-district might be to all of rural Bangladesh.

We have added a statement on generalizability in the discussion section [lines 319-322].

---

## [Editor Report · Decision Letter 2]

4 Jul 2023

Child marriage in rural Bangladesh and impact on obstetric complications and perinatal death: findings from a health and demographic surveillance system

PONE-D-22-26467R2

Dear Dr. Gurley Emily ,

We’re pleased to inform you that your manuscript has been judged scientifically suitable for publication and will be formally accepted for publication once it meets all outstanding technical requirements.

Kind regards,

Fadhlun Alwy Al-beity, MMed, PhD 

Academic Editor

PLOS ONE

---

## [Editor Report · Acceptance letter]

10 Jul 2023

PONE-D-22-26467R2 

Child marriage in rural Bangladesh and impact on obstetric complications and perinatal death: findings from a health and demographic surveillance system 

Dear Dr. Gurley:

I'm pleased to inform you that your manuscript has been deemed suitable for publication in PLOS ONE. Congratulations! Your manuscript is now with our production department. 

Kind regards, 

on behalf of

Dr. Fadhlun Alwy Al-beity 

Academic Editor

PLOS ONE